# High-Resolution Microstructure Characterization of Additively Manufactured X5CrNiCuNb17-4 Maraging Steel during Ex and In Situ Thermal Treatment

**DOI:** 10.3390/ma14247784

**Published:** 2021-12-16

**Authors:** Mihaela Albu, Bernd Panzirsch, Hartmuth Schröttner, Stefan Mitsche, Klaus Reichmann, Maria Cecilia Poletti, Gerald Kothleitner

**Affiliations:** 1Graz Centre for Electron Microscopy, Steyrergasse 17, 8010 Graz, Austria; gerald.kothleitner@felmi-zfe.at; 2ÖGI- Austrian Foundry Institute, Parkstraße 21, 8700 Leoben, Austria; bernd.panzirsch@ogi.at; 3Institute of Electron Microscopy and Nanoanalysis, TU Graz, Steyrergasse 17, 8010 Graz, Austria; hartmuth.schroettner@felmi-zfe.at (H.S.); stefan.mitsche@felmi-zfe.at (S.M.); 4Institute for Chemistry and Technology of Materials, TU Graz, Stremayrgasse 9, 8010 Graz, Austria; k.reichmann@tugraz.at; 5Institute of Materials Science, Joining and Forming, TU Graz, Kopernikusgasse 24, 8010 Graz, Austria; cecilia.poletti@tugraz.at

**Keywords:** additive manufacturing, microstructure, scanning transmission electron microscopy (STEM) in situ heating experiments

## Abstract

Powder and selective laser melting (SLM) additively manufactured parts of X5CrNiCuNb17-4 maraging steel were systematically investigated by electron microscopy to understand the relationship between the properties of the powder grains and the microstructure of the printed parts. We prove that satellites, irregularities and superficial oxidation of powder particles can be transformed into an advantage through the formation of nanoscale (AlMnSiTiCr) oxides in the matrix during the printing process. The nano-oxides showed extensive stability in terms of size, spherical morphology, chemical composition and crystallographic disorder upon in situ heating in the scanning transmission electron microscope up to 950 °C. Their presence thus indicates a potential for oxide-dispersive strengthening of this steel, which may be beneficial for creep resistance at elevated temperatures. The nucleation of copper clusters and their evolution into nanoparticles, and the precipitation of Ni and Cr particles upon in situ heating, have been systematically documented as well.

## 1. Introduction

Maraging steels are low-carbon, precipitation-hardenable martensitic steels with high strength and toughness, high temperature creep resistance and low temperature properties, but are also corrosion resistant. These properties qualify them as promising steels for structural applications and also as favorites for additive manufacturing due to their very good weldability. Few types of steels are actually used for additively manufactured components with complex geometries in aerospace, automobile, medical techniques, tooling and other industry applications: austenitic stainless steels (AlSl-316L/1.4404, AlSl-304/1.4307), duplex stainless steel (SAF2 705), maraging steels (17-4PH/1.4542, 15-5PH/1.4545, 18PH-300/1.2709), C-bearing (H13) tool steel and oxide dispersive steels (PM2000) [1,2,3,4,5]. The steels are selected based on application areas in which the additively manufactured components should have high corrosion resistance and long service lifetimes in extreme conditions, enhanced mechanical properties (strength, ductility, hardness, toughness, wear resistance, etc.), multivalent microstructural properties (from hard martensite to ductile multiphase components), different functionalities (electromagnetic or invar properties) and of course, affordable prices to make them appealing [5].

Most stainless and maraging steels produced by conventional methods use intermetallic compounds precipitated at temperatures between 400 and 650 °C as strengthening mechanism [6]. These precipitates with sizes between 50 nm (Cu precipitates) and 1 µm (Ni intermetallic phase) form inside the laths and at lath interfaces, and are expected to impede the movement of subgrains and dislocations (through the Orowan effect), thereby contributing to the precipitation hardening. However, the thermal stability of the precipitates at elevated temperatures is critical to resisting grain growth and dislocation recovery when long-term creep strength of the printed components needs to be achieved. The microstructure of an additively produced maraging steel in the as-built condition is refined due to the high cooling rates the material experiences, and consists of a few small retained austenite grains mixed with dislocation-rich ferritic/martensitic (bcc) laths. In addition, depending on the printing parameters and powder characteristics (e.g., uniformity of grain size and natural surface oxidation), element segregations and rather large oxide inclusions (TiO2, Al2O3 of about 10–20 µm, or other oxides containing Ti, Mo, Al and Si in various ratios) [4] have been observed at melt pool boundaries. Their presence, due to consistently high density of the part (absence of pores), is generally considered to have an unfavorable influence on the mechanical properties of the printed part [4,5,6,7] because of crack initiation. On the other hand, the in situ addition of dispersive oxide (ODS) powders during the additive production [8,9], or the use of environmental oxygen or nitrogen gases to form ODS (Ti_x_O_y_, Al_y_O_y_) or nitrogen dispersed particles (NDS) during printing, induces a significant increase in hardness and thermal stability [10]. However, such uniformly dispersed oxides with dimensions in the nanometer to micrometer range have been detected by only a few authors [11,12,13,14,15] in additively manufactured (L-PBF) stainless and maraging steels.

This work provides a detailed correlative microstructure study of X5CrNiCuNb17-4 steel [1,16], in powder form and additively manufactured by selective laser melting. This grade is a low-carbon, precipitation-hardened stainless steel with corrosion resistance improved by Cr and Ni, and the precipitation hardening is achieved by Cu addition and upon ageing heat treatment [1,16,17,18]. The gas atomized powder grains, the samples in the as-built condition and the sample that was ex situ heat treated according to the recommended scheme that also applies to the bar material (H1150 (620 °C): (P930)), were first investigated by advanced electron microscopy and nanoindentation.

Further, we performed in situ thermal treatment with a high-resolution scanning transmission electron microscope (STEM) and correlated the results with simultaneous thermal analysis (STA) measurements in order to understand the relationship between the powder characteristics and the microstructural details that might influence the properties of the printed parts. In particular, we systematically followed the evolution of the nanometric spherical oxides that formed during printing, and the chromium and copper precipitate nucleation and coarsening upon heating of the sample in the as-built condition up to 950 °C. Such experiments have, to the best of our knowledge, only been reported for AlSi10Mg alloy [19].

## 2. Materials and Methods

The microstructure of X5CrNiCuNb17-4 steel in powder form and additively manufactured in as-built condition, and the thermally treated specimen (1040 °C/30 min/air cooling + 620 °C/4 h/air cooling), have been extensively investigated by optical, scanning and high-resolution transmission electron microscopy techniques and simultaneous thermal analysis. For both samples we measured the hardness (replacing the tensile test) and determined the fatigue strength via nanoindentation technique. 

The spherical powder was produced by argon gas atomization and had a broad particle size distribution between 14 and 45 µm (D10 (μm) 18–24, D50 (μm) 29–35, D90 (μm) 42–50) [16,20]. The cycled powder (mixture of pristine and the excess of a previous printing) was used for additive manufacturing by selective laser melting. The chemical compositions of the powder and of the additively manufactured sample in the as-built condition, as determined by wet chemical analysis, are listed in Table 1. The chemical composition of the as-built sample differed slightly from that of the powder, indicating that some elements (C, Si, Mn, Nb, Mo) were evaporated during the printing process. After thermal treatment, the chemical composition remained unchanged. The printing parameters for the as-built sample are given in Table 2.

The X5CrNiCuNb17-4 powder, and the as-built and thermally treated samples, were first examined in a scanning electron microscope (SEM, Zeiss Ultra 55, Carl Zeiss AG, Oberkochen, Germany). An adhesive tape was used to pick up the powder grains for morphology and sphericity analysis. A second batch of the powder was embedded in a resin (CaldoFix-Struers), polished down to a few tenths of a micrometer and thinned by Ar-ion milling at cryogenic temperature. The additively manufactured samples were cut and polished along the build direction (Z-axis) and the basal plane (XY-plane), using diamond and aluminum suspensions as final polishing steps for electron backscatter diffraction (EBSD) measurements. Electron backscatter diffraction (EBSD) measurements were carried out on a Zeiss Ultra 55 scanning electron microscope equipped with a Thorlabs high-resolution CCD camera and the TSL OIM DC software V7.3 from Ametek (Berwyn, PA, USA). An acceleration voltage of 20 kV and a beam current of 12 nA were used for these EBSD measurements. Data evaluation was done with the help of the TSL OIM Analysis software V8.0 from Ametek. Two different areas were analyzed: 150 µm × 150 µm with a step size of 150 nm and 50 µm × 50 µm with a step size of 50 nm. With these step sizes and a restriction that a grain has to include at least 6 points a minimum detectable grain size of 410 and 140 nm respectively is achievable. On all scans a grain dilation was applied to reduce the noise in the measurement (change of points less than 4%). The grain tolerance angle was set to 5°. SEM micrographs were acquired using backscatter, secondary electron, and in-lens detectors. We used energy dispersive X-ray spectrometry for the analytical studies (Oxford Si(Li) detector) and EBSD for crystallographic studies of the grains and grain boundaries.

The ex situ samples for scanning transmission electron microscopy (STEM, Thermo Fischer Scientific, Waltham, MA, USA) analysis were prepared by Ar ion milling at cryogenic temperature, while for the in situ heating experiments, a focused ion beam (FIB) prepared lamella, cut from the as-built sample, was mounted on a MEMS heating chip (DENS solutions-Wildfire H+ DT, heating/quenching rate 200 °C/ms and settling time 2 s). For the high-resolution STEM studies, we used a probe-corrected microscope FEI Titan G3 60-300 (Thermo Fisher Scientific, Waltham, MA, USA) operated at 300 kV. HR-STEM micrographs were acquired using the annular (ADF) and high angle annular dark field (HAADF) detectors. Analytical studies were performed using energy dispersive X-ray analysis (EDS) and electron energy loss spectroscopies. For EDS the microscope was equipped with a windowless silicon drift detector - Super-X (Chemi-STEM technology, Thermo Fisher Scientific, Waltham, MA, USA), and for electron energy loss spectroscopy with a dual EELS Quantum Gatan Imaging Filter (GIF, Gatan Inc., Pleasanton, CA, USA).

The nanoindentation measurements were performed using the NHT3 Nanointender from Anton Paar (Graz, Austria). During the measurements, the applied force and displacement are measured digitally and then used for the evaluation of the material’s hardness. The macroscopic hardness of the stepped plate was determined by Vickers hardness measurements (HV1) while applying a force of 10 N. For the microscopic hardness, we applied a force of 4 mN over 245 indentation points. The samples were tested three times in the cross-section (YZ plane) in each step.

Simultaneous thermal analyses (STA) combining thermogravimetry (TG) and differential scanning calorimetry (DSC) were performed in a NETZSCH STA 449F1 Jupiter (NETZSCH-Gerätebau GmbH, Selb, Germany) in an argon atmosphere, applying different heating and cooling rates.

## 3. Results

The morphology, chemical composition, particle size distribution and roundness factor of X5CrNiCuNb17-4 powder grains have been studied in detail and published elsewhere [1,20]. An overview SEM image of the as-received powder is presented in Appendix A.

We observed a rather large size distribution and a large number of satellites. The images in Figure 1 are from a chosen irregular grain in the as-received powder. The irregular grain has a grain core and accretionary forms on top of it. The apparent interlamellar pores are at the interface between the grain body and the accretionary envelope. The formation of the accretionary form in this case was probably due to the collision of two powder grains, one of them being not entirely solidified at the moment of collision. STEM analysis of the focused ion beam lamella cut from such a grain with accretionary forms (Figure 1a,b) revealed the presence of a region with slightly increased Ni content (from 3.5 at. % to 4.8 at. %) (Figure 1c, #1). Figure 1d shows the high-resolution image from the region indicated in #1 and the FFTs from calculated matrix #4 and Ni enriched phase #5, respectively. The FFT image from #5 do not differ from #4, which indicates a martensite/ferrite phase (Miller indices (101) and 200), except for some stress that appears as a diffuse signal. Figure 1e shows the EDX spectrum from #5 with the following concentrations: Cr—14.4 ± 1.2 at. %; Fe—73.4 ± 5.2 at. %; Ni—4.8 ± 0.5 at. %; Cu—7.4 ± 0.9 at. %.

A Cu-metallic layer (Figure 1c, #2) and a nm-thin layer of amorphous carbon were observed at the interface between the particle body and the accretionary envelope (Figure 1c, #3). Furthermore a 10 nm thin passive oxide layer was found to cover the particles and irregularities.

The geometry of the printed samples was stepwise (Figure 2a) to determine the effect of redundant heat during the printing process on the microstructure and mechanical properties. The mechanical properties for the as-built sample and the thermally treated sample (1050 °C/3 min/air + 620 °C/4 h/air), measured by nanoindentation, are presented in Figure 2b. The measurement was performed in the YZ plane for all three areas and showed a low influence of the thermal treatment on the microhardness.

The as-built sample was extensively investigated in the build-up direction (*Z*-axis) and in the basal plane (XY-plane) by means of optical microscopy and scanning electron microscopy (Figure 3). The SEM images show distinct areas of microsegregation, which can be attributed to the quick cooling and solidification during the laser process. The microsegregations occurred in lines at the interface between the melt pools—following the layers—and had a reduced hardness compared to the matrix, so they could serve as preferential crack paths. Their chemical analyses by EDX showed a mixture of Ni, Mo, Ti, Nb and W with various concentrations, most of them having high concentrations of Mo and Ni. Some examples of such segregations with EDX analysis and the respective elemental quantifications are shown in Appendix A.

EBSD analysis showed a fine microstructure with small austenite grains of 280 nm up to 1.5 µm in size, mixed with larger ferrite and martensite grains of about 10 µm (Figure 3g,h at lower magnification for a better overview and Figure 3i,j at higher magnification to highlight the small retained austenite grains). The austenite fraction, as quantified from the EBSD images in both Z-direction and XY-plane, ranged from 2% to 3.1%. No clear columnar grains of increased size were observed in the heat-affected zones at the boundaries of the melt pools, indicating that the printing parameters were optimized for the best homogeneous microstructure that a sample can have in the as-built condition.

At higher magnifications, we noticed micrometer-sized pores and dark or bright small spherical features. The chemical analysis of the EDX spectra in Figure 3f #2 shows the following elements: Si—0.24 wt.%, Mo—4.09 wt.%, Cr—5.01 wt.%, Mn—0.34 wt.%, Fe—69.02 wt.%, Co—7.17 wt.%, Ni—11.94 wt.% and Cu—1.27 wt.% (systematic error of the EDX quantification in SEM is ~10%).

STEM analysis (Figure 4a–c) showed the dark spherical features that decorate the matrix to be distributed inside the martensite or ferrite grains, but also at the grain boundaries. Furthermore, the high-resolution STEM image of such a particle in Figure 4d and the X-ray elemental maps and line scan in Figure 4e, confirmed that they are indeed nanometric oxide phases containing Si, Al, Ti, Cr and Mn. Their internal structure is disordered crystalline and core-shell-like, with a core consisting mainly of silicon and a shell of titanium and chromium, while aluminum and manganese are evenly distributed. However, in some nano-oxides, the shell of titanium and chromium was found to be sheet like (probably a couple of atomic layers thick) and crystalline. While the other elements were expected, the presence of Al and Ti atoms probably originated from the alloy ingot casting, which was subsequently converted into powder. However, their concentrations seem to be high enough to form densely dispersed oxides in the matrix.

The thermally treated sample (1050 °C/3 min/air + 620 °C/4 h/air) (Figure 5a,b) exhibited a refined microstructure. The segregations (Figure 5c with EDX #1 and Appendix A) at the melt pool interfaces were not entirely dissolved, however. The Nb-rich carbides and the nano-oxides (Figure 5d,e with EDX #2 and #3) were also still preserved. As expected, the intermetallic Ni precipitates (200–700 nm) formed mainly at the grain or martensite lath boundaries.

Figure 6 and Figure 7 present STEM HAADF images and EDX maps for areas containing both Ni-precipitates and nanoparticles (oxides and carbides) at low and high magnification, respectively. Along with the Nb-rich and nano-oxides that did not change their morphologies or chemical compositions, we found a high density of Cu nanoparticles (5–20 nm) uniformly distributed in the matrix. The element concentrations in the oxides and carbides varied only slightly and can thus be considered stable phases. Exemplar line scans for the typical nanoparticles indicated by arrows on the Mn, Ti and Nb maps in Figure 6 are also shown. Both nanoparticle types, nano-oxides and nano-carbides, were not affected at all by the heat treatment at 620 °C.

The chemical composition of the matrix as extracted from the EDX spectrum image that also served as source for the EDX maps showed concentrations of the major elements as follows: Fe: 77.31 ± 13.0 wt.%, Cr: 16.26 ± 1.9 wt.%, Ni: 3.37 ± 0.6 wt.% and Cu: 3.06 ± 0.5 wt.%. The Nb rich particles contained: C: 5.6 ± 0.3 wt.% and Nb: 39.64 ± 6.1 wt.%; and the contributions from the matrix underneath were Fe: 37.22 ± 6.2 wt.%, Cr: 9.8 ± 1.5 wt.%, Ni: 2.68 ± 0.5 wt.% and Cu: 5.09 ± 0.8 wt.%. The Ni intermetallic phase contained Ni: 11.02 ± 1.2 wt.%, the contributions from the matrix being Fe: 71.13 ± 12.2 wt.%, Cr: 14.91 ± 1.8 wt.% and Cu: 3.69 ± 0.6 wt.%. After normalization to the matrix and analysis of the FFT image calculated from the high-resolution STEM image presented in Figure 6c, we found the Ni phase to be a µ-type (A_6_B_7_).

Nevertheless, possible nanopores partially filled with oxygen and having Cu, Mn and Ni adsorbed on their walls are also shown in Figure 7b. Their presence, however, might have been induced by the electron beam affecting the nano-oxides situated closed to the surface of the very thin STEM sample.

In order to clarify whether the oxides are stable at elevated temperatures and to observe the nucleation of the nanosized Cu, Cr and Ni precipitates, we performed simultaneous thermal analysis (STA) measurements of the as-built sample, and STEM in situ heating experiments accompanied by EDX analyses. 

In the first STA experiment, the heat treatment was simulated with a heating rate of 10 K/min up to 1050 °C and a dwell time of 30 min, followed by a rapid cooling with 50 K/min, all in an argon atmosphere (Figure 8). The STA curve exhibits an exothermal peak at 483 °C in the first heating segment, followed by another broad exothermal peak above 600 °C. In the cooling segment, no signals were detected that would indicate reversibility for these processes. The measurement at a higher heating rate 35 K/min (not shown here) showed a shift in the first exothermal peak toward higher temperatures (501 °C). 

In addition, the as-built sample was in situ heated in the high-resolution microscope operated in STEM mode, starting at room temperature (23 °C) and heated to 950 °C at a rate of 8.6 K/min between 23 and 550 °C and at a rate of 2.8 K/min between 550 and 950 °C (Appendix A). It is worth noting that heating at a temperature above 950 °C was hindered by the fact that the matrix in the very thin lamella decomposed almost completely (Figure 9a).

The overall appearance of the microstructure in the HAADF images seemed to remain stable up to 600 °C (Figure 9a), even though the STA measurement indicated an exothermic peak at 480 °C, which is related to the formation of Cu clusters and nanoparticles [22]. Nevertheless, on the high magnification EDX maps taken at 500 °C, we could indeed observe a few Cu nanoparticles (Cu map and line scan on Figure 10). In the images acquired with the HAADF detector, however, the formation of the nanoparticles and their continuous coarsening (intermetallic Cu, Cr and Ni) were first observed above 650 °C and up to 950 °C. (At this point we have to note that imaging alone, also using different detectors, without analytical evidence, did not deliver the complete information about the dynamics of the alloy system.) While the coarsening of Cu and Ni precipitates was rather slow, the coarsening rate for Cr precipitates proved to be higher.

The nano-oxides, on the other hand, remained stable up to 950 °C and retained their spherical morphology; complex or core-shell-like chemical composition; and crystallographic disorder. Figure 9b shows images at temperatures between 23 and 760 °C from an area at high magnification containing few nano-oxides. Above 700 °C, the matrix in this region started to decompose quickly, covering some oxides. However, they are still visible at a lower magnification in Figure 9a. Chemical analysis of a nano-oxide was performed at room temperature and compared to the analyses at 500 and 800 °C. We did not detect any changes on the EDX maps shown in Figure 10 at any of the three temperatures. However, the EDX maps at 800 °C were recorded at lower magnification to feature the changes undergone by the matrix surrounding the nano-oxide. The high-resolution STEM images of such a particle at different temperatures presented in Appendix A show the copper segregation in the proximity of the nano-oxide at temperatures above 600 °C. The shape of the particle was slightly changed at 850 °C due to the Cu and Cr segregations and matrix deterioration.

Figure 11 exemplifies the STEM HAADF image and chemical analyses of a large area at 700 °C. The EDX maps display the coarsened Cr (about 200 nm) and Ni (larger than 250 nm) precipitates; the Cu (about 20–30 nm) and oxide nanoparticles; but also silicon segregations of about 2–5 nm at martensite lath interfaces. A particle containing both carbon and chromium can be observed in the related EDX maps. The chemical analysis shown in the line scan presented in Figure 11 corresponds to the M_23_C_6_ phase. The observed precipitation of M_23_C_6_ is contrary to the general knowledge that C-containing maraging steels’ nucleation should be restricted due to the low carbon concentration. Phase diagrams for PH-17 grade [23,24] show a very low mass fraction for the M_23_C_6_ phase with possible coarsening above 650 °C. We assume that the nucleation of the M_23_C_6_ precipitate during in situ heating was due to accelerated surface diffusion of Cr and C atoms at higher temperatures. Some C atoms were also probably supplied by the fixing alloy (mixture of Pt and C) for the FIB lamella on the heating chip. Although we tried to minimize the contribution of the fixing alloy by preparing the FIB lamella in a window-like manner (thin middle area with very thick edges acting as barriers for diffusion), this effect could eventually not be completely prevented.

When comparing the STA measurements with the in situ STEM measurements at 450 and 500 °C, we observed that the exothermic peak present at 481 °C (10 K/min) or 501 °C (35 K/min) did not resemble any dramatic changes in the microstructure that could be detected on the images at low magnification. At higher magnification, however, the Cu-EDX map (Figure 10) demonstrates the formation of dense Cu clusters and nanoparticles, as reported in the literature for pH 15–5 [21]. This process is probably driven by the continuous stress release that starts already at low temperatures (probably above 200 °C), and pipe diffusion through dislocations. Besides pipe diffusion, the surface diffusion accelerated the growing of the Cu nanoparticles at higher temperatures and the nucleation of Cr and Ni intermetallic precipitates, followed by their coarsening that gives rise to the broad exothermic peak starting at 620 °C. 

Therefore, we can understand the STA curve by considering matrix relaxation at low temperatures (below 450 °C), the formation of Cu clusters between 470 and 520 °C, followed by coarsening until 950 °C and precipitation and coarsening of Cr- and Ni particles between 620 and 950 °C. The nanosized oxides remained unchanged over the whole range of temperatures; therefore, no influence on the STA curve was expected.

## 4. Discussion

Powder analysis confirmed the thin passive oxide layer covering the particles and irregularities, apparently supplying the system with sufficient oxygen to ensure the formation of nano-oxides during the printing process. The nano-oxides in the final sample may have a high density near the melt pools or be uniformly distributed throughout the matrix, depending on the printing parameters. In the thermally treated sample, the nano-oxides were not affected in any way by the heat treatment at 620 °C (also acknowledged by Sun et al. (2018) [7]). Furthermore, the in situ heating experiment up to 950 °C confirmed their stability in all aspects (Figure 11), suggesting that they could be beneficial in terms of strength or creep resistance at elevated temperatures.

The oxides can often be misinterpreted as pores, since micrometer-sized pores with different morphologies and nanopores are present in additively manufactured parts. While the micrometer-sized pores are easily detected by different microscopic methods, such as optical microscopy, SEM and X-ray computer tomography, the detection of nanopores requires high-resolution investigation methods, such as STEM, accompanied by chemical composition analyses using X-ray and electron energy loss spectroscopy (EELS). Advanced methods as the EELS fine structure (ELNES) analysis of the light element edges can even distinguish between molecular oxygen (or nitrogen) in gas form and in an amorphous or crystalline state. Furthermore, this method can be applied for the detection of such atoms at grain or melt-pool boundaries. It is also worth noting that XRD analyses or other diffraction techniques do not provide information about such nano-oxides, as they are generally amorphous.

Furthermore, the amorphous carbon located at the interfaces between the powder particle body and the envelopes (irregular particles), in addition to the nominal concentration given in Table 1, supplied sufficient carbon for the formation of NbC, but also a few chromium carbides at higher temperatures, as presented in Figure 9 and Figure 11. The NbC particles appear on the EDX maps (Figure 6 and Figure 7) and with bright contrast on the SEM images (Figure 5e, #2). The chromium carbides are only visible on the HAADF images taken above 660 °C (Figure 9b) and on the EDX maps corresponding to 700 °C in Figure 11.

The STA curve in Figure 8 confirms our results for the in-situ heating measurements and validates this method for the heat treatment studies on other steel grades and alloys. 

The measured Vicker hardness of the additively manufactured sample lies between 360 and 380 HV1, which is in line with other data presented in the literature. However, recent data [25] show that the ductility is lower than for samples prepared by conventional metallurgy, and this can probably also be attributed to the presence of the oxides and carbides. Further, the heterogenous distribution of the elements that segregate at the melt-pool boundaries can also be responsible for crack initiation.

## 5. Conclusions

Correlative electron microscopy and STA investigations were performed for X5CrNiCuNb17-4 maraging steel to better understand the microstructural characteristics of the final additively manufactured parts and their evolution upon in situ heating up to 950 °C.

The high-resolution microstructure analyses of irregular powder grains revealed the thin copper segregations and an amorphous carbon layer at the interface of the grain body and accretion shell, and the surface oxidation of about 10 nm of the powder grains.

Further, the microstructure characterization of the as-built condition unveiled the presence of retained austenite and nanosized oxides distributed in the matrix. These particles have been shown to persist in the thermally treated state, and together with the formation of the intermetallic Ni precipitates at the interfaces between the laths and the Cu nanoparticles, might contribute to the improved hardness of the material. 

During in situ heating, by STEM we observed the formation and coarsening of Cu nanoparticles that reached dimensions larger than 200 nm at 700 °C, and precipitation of Ni and Cr particles that continuously coarsened at higher temperatures, reaching 500 nm at 950 °C. These observations are in line with the STA measurements at higher temperatures and investigations presented in the literature for bar materials. However, a comparison between AM and conventionally produced samples provided by [24] showed that the densities of Cu nanoprecipitates and their dimensions are relatively higher in AM samples. Therefore, we consider that the higher stress due to fast cooling, which leads to high dislocation densities, favors the nucleation of copper clusters through pipe diffusion at temperatures above 450 °C and enables their fast coarsening at higher temperatures.

The origin of the core-shell nanosized oxides already observed in the as-built sample is most probably linked to the very thin superficial oxidation layer on the powder particles. The in situ heating experiment in the STEM confirmed their exceptional stability up to 950 °C. They preserved their dimensions, spherical morphology and chemical composition, and might be considered as strengthening particles in addition to Ni and Cu precipitates.

Consequently, we conclude that the controlled superficial oxidation of steel powder grains may in fact be reversed into an advantage, to produce oxide-dispersive, strengthened, additively manufactured steel parts with higher thermal stability.

## Figures and Tables

**Figure 1 materials-14-07784-f001:**
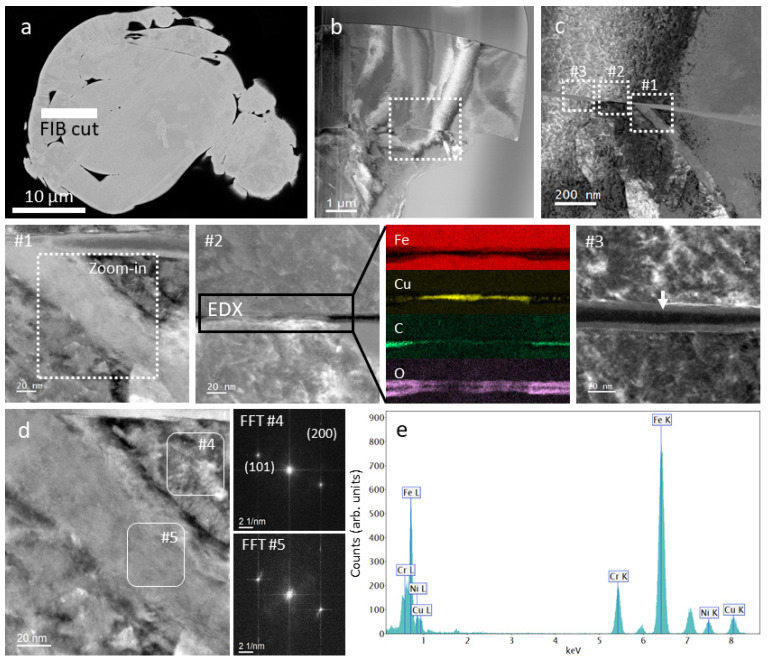
Powder characterization by SEM and STEM. (**a**) SEM of a corrupted particle with the FIB lamella position; (**b**) STEM ADF image of the FIB lamella; (**c**) STEM HAADF image at higher magnification. The insets (**#1**), (**#2**) and (**#3**) represent different regions at the interface of the particle body with the envelope. (#**1**)—Ni enriched phase (inset depicts the zoom-in for (**d**), (**#2**)—Cu metallic interface and (#**3**)—amorphous C and iron oxides. The Fe, Cu, C and O EDX maps reveal the interface from area (**#2**). (**d**) High-resolution STEM ADF image with insets defining the matrix (**#4**) and the Ni enriched region (**#5**) for FFT analysis; (**e**) EDX spectrum from (**#5**): Cr—14.4 ± 1.2 at. %, Fe—73.4 ± 5.2 at. %, Ni—4.8 ± 0.5 at. %, Cu—7.4 ± 0.9 at. %.

**Figure 2 materials-14-07784-f002:**
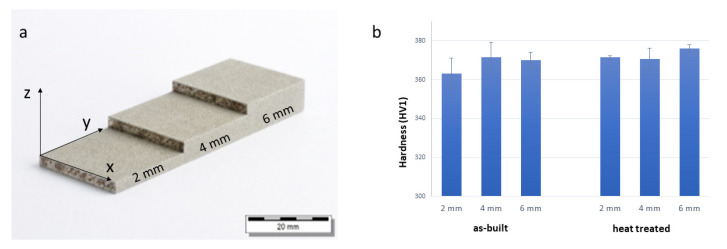
(**a**) Image of the printed sample with the step-wise geometry; (**b**) nanoindentation measurements performed in the cross-section (YZ plane) of the as-built and heat-treated samples for the three areas with different thicknesses (2, 4 and 6 mm).

**Figure 3 materials-14-07784-f003:**
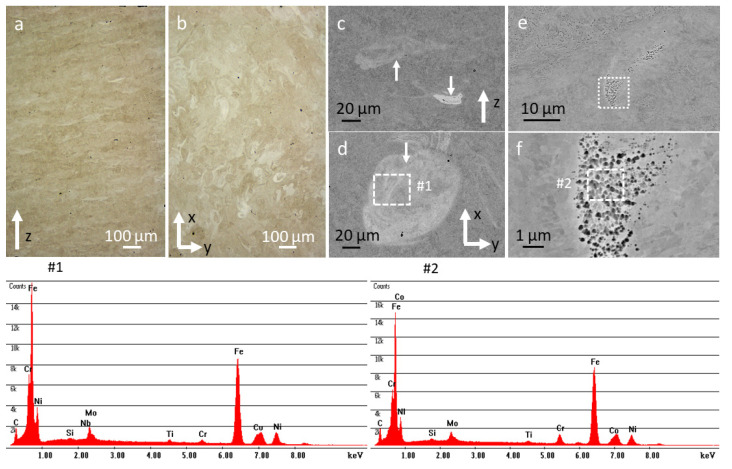
As-built sample characterization by optical microscopy (infinite focus microscopy) and SEM. (**a**) IFM of the sample polished in the build direction (Z-axis) and (**b**) in the basal plane; (**c**,**d**) SEM of the as-built sample in the build direction (Z-axis) and basal plane with the EDX spectrum (**#1**) from a microsegregation area; (**e**,**f**) SEM images with the EDX spectrum (**#2**) from a region with pores and spherical nano-sized features; and inverse polfigure maps of the as-built sample along (**g**) the Z-axis and (**i**) basal plane for ferrite-martensite, and (**h**–**j**) details of the austenite phases of (**g**) and (**i**), respectively; micro bars for the inverse pole-figure maps: 50 and 15 µm.

**Figure 4 materials-14-07784-f004:**
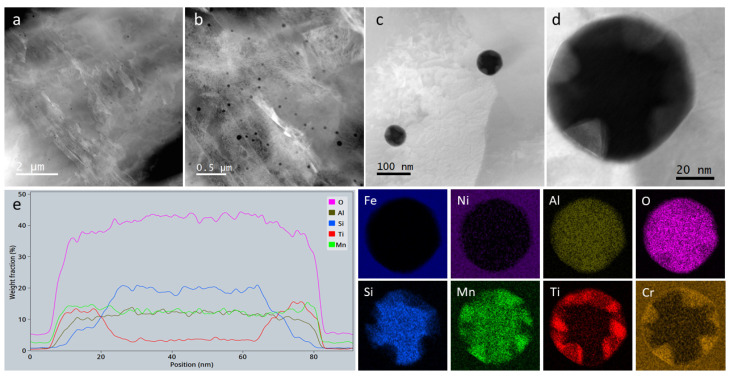
STEM investigations for the as-built sample. (**a**,**b**) Low magnification HAADF images; (**c**,**d**) high resolution HAADF images; (**e**) EDX line scan and EDX maps from the particle represented in (**d**).

**Figure 5 materials-14-07784-f005:**
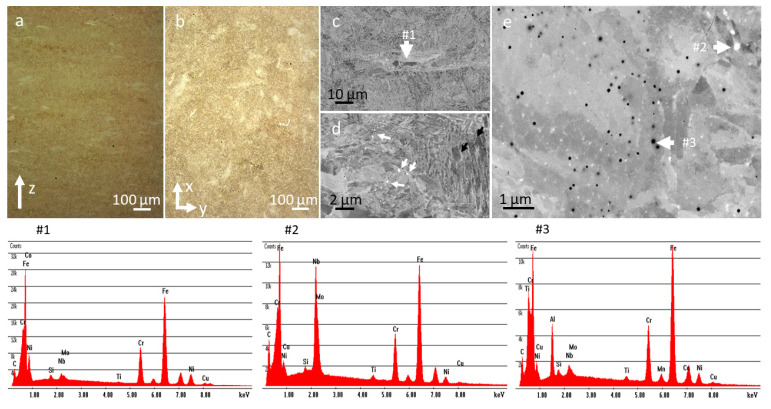
Thermally treated sample (1050 °C/3 min/air + 620 °C/4 h/air) characterization by optical microscopy and SEM. (**a**,**b**) IFM images in the Z-axis and basal plane; (**c**,**d**) SEM images at low magnification with the EDX spectrum from the segregation area (**#1**); and (**e**) higher magnification SEM image with the EDX spectra from the bright (**#2**) and dark (**#3**) particles.

**Figure 6 materials-14-07784-f006:**
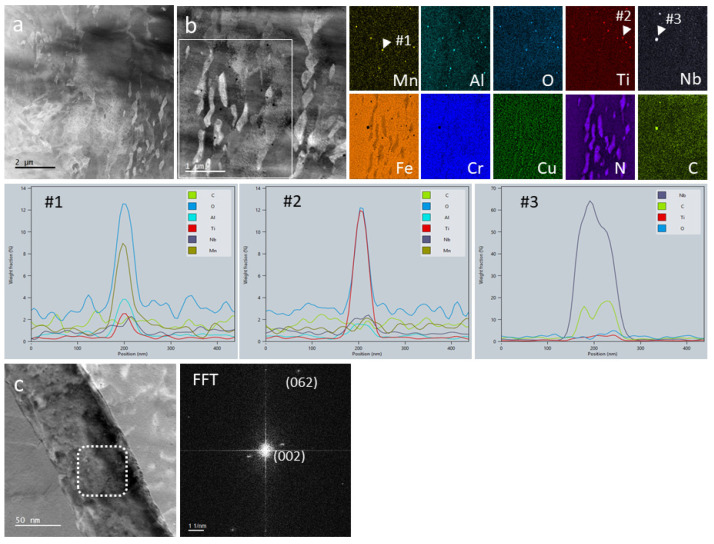
STEM investigations for the thermally treated sample. (**a**) HAADF image at low magnification. (**b**) HAADF image with the rectangle indicating the area for EDX maps. EDX line scans over particles (**#1**) (manganese oxide), (**#2**) (titanium oxide), (**#3**) (niobium carbide). (**c**) ADF STEM image and FFT analysis of the Ni-µ phase [21].

**Figure 7 materials-14-07784-f007:**
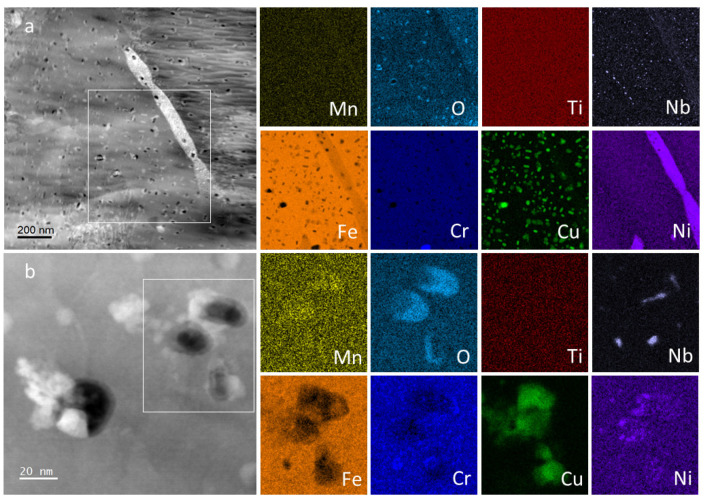
(**a**) EDX maps from an area containing nanometric particles in a thermally treated sample. (**b**) EDX maps at higher magnifications to reveal nanometric pores filed with oxygen and having Cu, Mn and Ni adsorbed on the walls.

**Figure 8 materials-14-07784-f008:**
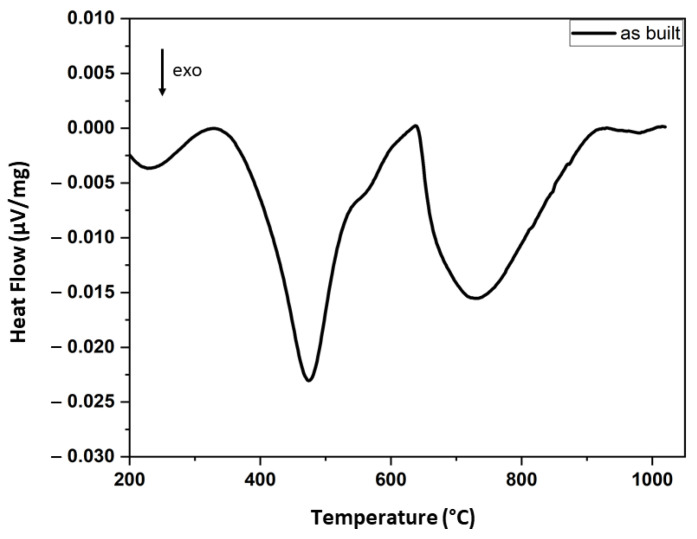
Simultaneous thermal analysis (STA—extracted DSC curve) measurements of the as-built sample at a 10 K/min heating rate.

**Figure 9 materials-14-07784-f009:**
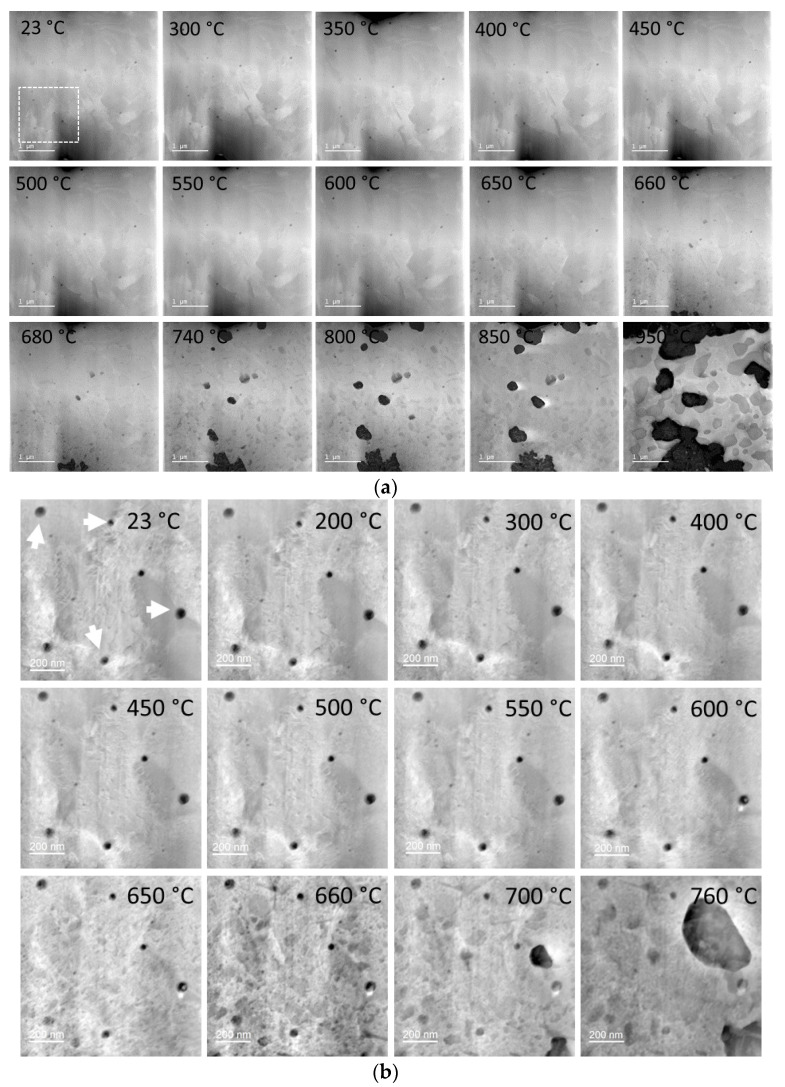
In situ heating study for the as-built sample (**a**) at a lower magnification to observe the nucleation and coarsening of the Cr- and Ni-rich precipitates, and (**b**) at a higher magnification to follow the evolution of the nanometer-sized oxides.

**Figure 10 materials-14-07784-f010:**
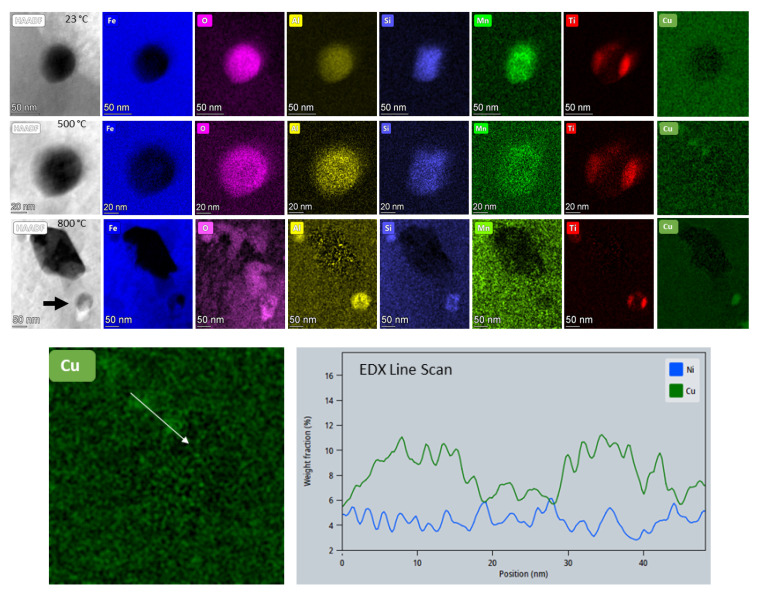
EDX analysis of a nanometer-sized oxide in the as-built sample at room temperature (23, 500, and 800 °C). EDX line scan over two Cu nano-precipitates at 500 °C (Ni signal on the line scan is displayed for comparison).

**Figure 11 materials-14-07784-f011:**
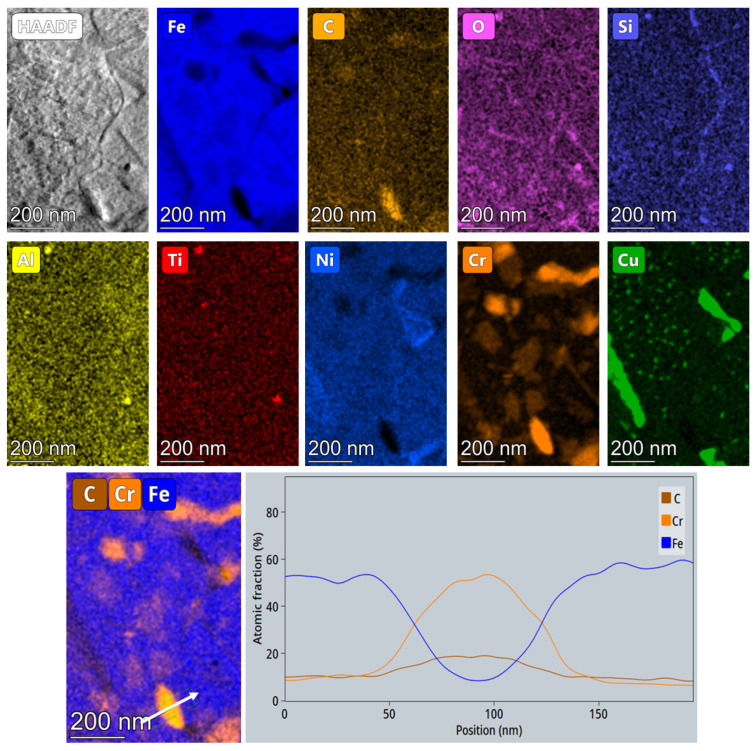
EDX analysis at lower magnification after heating at 700 °C. Image and EDX line scan over the M_23_C_6_ particle.

**Table 1 materials-14-07784-t001:** Chemical composition of the powder [1,16] and the as-built sample (wt.%). The standard deviations of the measurements lie between 3% and 5%.

wt.%	C	Si	Mn	Cr	Ni	Cu	Nb	P	S	Mo	Fe
Powder [1]	0.07	0.7	1.0	15–17	3–5	3–5	0.45	0.025	0.015	0.5	bal.
As-built	0.04	0.34	0.36	15.2	4.80	3.03	0.25	0.02	0.006	0.26	bal.

**Table 2 materials-14-07784-t002:** Printing parameters of X5CrNiCuNb17-4 steel.

Power	Speed	Hatch Distance	Layer Thickness
250 W	850 mm/s	0.1 mm	0.05 mm

## Data Availability

Data sharing is not applicable to this article.

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
