# Peer review of "High-Resolution Microstructure Characterization of Additively Manufactured X5CrNiCuNb17-4 Maraging Steel during Ex and In Situ Thermal Treatment"

_materials, 2021, doi:10.3390/ma14247784_

Round 1
Reviewer 1 Report
The article entitle "High-resolution microstructural characterization of additively manufactured X5CrNiCuNb17-4 maraging steel during ex and in situ thermal treatment" is focused on microstructural characterization of powder and SLM additively manufactured parts of X5CrNiCuNb17-4 maraging steel to understand the relationship between the properties of the powder grains and the microstructure of the printed parts.
Presented research is interesting and sufficiently novel. Scope of manuscript is original.
Experimental details are presented in such a manner to enable other researcher to reproduce the applied methodology.
Presented results are very well presented, discussed and supported by previously published literature data.
In conclusion, this paper is presenting interesting and scientifically sound results which should be published in Materials.
Author Response
Thank you very much for the review and appreciation!
Reviewer 2 Report
General note: Paper does not contain results of electron diffraction patterns, that are necessary to accept the article. This is the main reason for rejection. Other remarks to correction:
- What are the novelty aspects of the paper?
- There is no confirmation of the presence of Ni-intermetallic phases in Fig. 1.
- Marker has to be added to the map presenting the elemental distribution in Fig. 1
- EBSD methodology is not described. It is necessary to provide more information. What about the step size and area of EBSD analysis? What about post-processing of EBSD data? What software was applied? What kind of clean-up procedures were applied? How the grain was defined? What about the minimum grain size and grain tolerance angle?
- Markers in EBSD maps are illegible (Fig. 3),
- Microareas for austenite testing using EBSD are too small.Only points can be visible to be the result of noise – Fig. 3g, i. These results are very doubtful.
- There are no results of phase compositions of analysed particles. Only EDS measurements were performed.
- What does it mean “martensitic refined microstructure”?
- The EDS results of matrix should be added to the Fig. 5. A discussion should be carried out regarding differences in elements occurring in particles and matrix.Due to the specifics EDS method, the results of EDS shown in Fig. 5 come from matrix, as well.
- Where are the results of NbC phase composition? There are no electron diffraction patterns.
- “Moreover, we also observed the precipitation of Cr23C6…”. Where are these results?
- The manuscript includes errors. For instance: “cupper segregation”.
- The “mechanical properties of the final additively manufactured parts” are mentioned in the conclusions, but neither one of them applies to it.
- Conclusions should highlight the new findings of the paper in comparison to the state of the art.
Author Response
General note: Paper does not contain results of electron diffraction patterns, that are necessary to accept the article. This is the main reason for rejection. Other remarks to correction:
Answer:
Thank you very much for the review and for the comments that have helped us to greatly improve this paper. We addressed all points and provided explanations and the corresponding changes in text as well as additional figures in the supplementary material.
Conventional electron microscopy includes in most cases SEM and TEM imaging and electron diffraction patterns. In some cases, EDX measurements with conventional detectors are also included.
In order to address relevant information related to the nano-oxide particles we went beyond conventional TEM and performed advanced electron microscopy investigations. It means we performed high resolution scanning transmission electron microscopy (HRSTEM) coupled with EDX using a novel detection system containing 4 windowless SSD detectors, which means that we can also measure and quantify light elements. We used Spectrum Imaging for the EDX measurements, therefore in each pixel of investigated area we have a spectrum from which we can extract single information or quantified elemental maps. Concerning diffraction patterns: reliable diffraction information for such small complex particles can only be achieved by nano-diffraction. We did not perform such kind of investigations since our first goal was to demonstrate the stability of the nano-oxides formed during the 3D printing process at higher temperatures.
However, HR images have been acquired for the as-built and thermally treated samples and the fast Fourier transformed images (FFT) – that enable the measurement of the lattice distances (Millers indices) – have been calculated. In order to keep the present paper at a reasonable length presenting only relevant information we did not include this analysis in the main text but we added it in the supplementary material during the review. Moreover, electron diffraction (SAD) for a regular powder particle has been already published in [20].
- What are the novelty aspects of the paper?
This paper is the first attempt to correlate the powder characteristics to the microstructure of 3D printed parts using this kind of alloy. Since the microstructure of the as-built parts differ essentially from their forged counterparts due to the fast cooling.
The presence of the oxide layer on the surface of the particles that induces the formation of the nano-oxides during printing, thus enhancing the mechanical properties (hardness) at higher temperatures, has not been documented and in detail investigated so far.
The introduction has been improved.
2. There is no confirmation of the presence of Ni-intermetallic phases in Fig. 1.
Answer:
Thank you for bringing this to my attention. I have carefully checked the EDX and HR images data again and indeed there is no Ni-intermetallic phase, the Ni content is however slightly increased in this region from 3.5 at% in the matrix to 4.8 at%. The HR images indicate a stressed lattice but the interatomic distances in the given orientation do not differ from the matrix.
The main text on page 4 has been corrected:
“STEM analysis of the focused ion beam lamella cut from such a grain with satellites and accretionary forms (Fig. 1a and b) revealed the presence of a region with slightly increased Ni content (from 3.5 at% to 4.8 at%) (Fig. 1c #1 and suppl. Fig. 2)…..”
Suppl Fig 2 has been added to supplementary information. It presents the FFT images from the matrix and enriched Ni region as well as the respective EDX spectrum.
3. Marker has to be added to the map presenting the elemental distribution in Fig. 1
Answer:
The markers were present, however in white. The color has been changed in black.
4. EBSD methodology is not described. It is necessary to provide more information. What about the step size and area of EBSD analysis? What about post-processing of EBSD data? What software was applied? What kind of clean-up procedures were applied? How the grain was defined? What about the minimum grain size and grain tolerance angle?
Answer:
The following text has been introduced in the Materials & Methods section on page 3:
“Electron backscatter diffraction (EBSD) measurements were carried out on a Zeiss Ultra 55 scanning electron microscope equipped wih a Thorlabs high-resolution CCD camera and the TSL OIM DC software V7.3 from Ametek. An acceleration voltage of 20kV and a beam current of 12nA were used for these EBSD measurements. Data evaluation was done with the help of the TSL OIM Analysis software V8.0 from Ametek. Two different areas were analyzed: 150x150 µm with a step size of 150 nm and 50x50 µm with a step size of 50 nm. With these step sizes and a restriction that a grain has to include at least 6 points a minimum detectable grain size of 410 nm and 140 nm respectively is achievable. On all scans a grain dilation was applied to reduce the noise in the measurement (change of points less than 4%). The grain tolerance angle was set to 5°.“
5. Markers in EBSD maps are illegible (Fig. 3),
Answer:
The markers have been improved and the scale has also bin indicated in the capture of Fig. 3.
6. Microareas for austenite testing using EBSD are too small. Only points can be visible to be the result of noise – Fig. 3g, i. These results are very doubtful.
Answer: -
The austenite EBSD images at lower magnification have been replaced with images at higher magnification and the zoomed area is indicated in Fig. 3 g and i.
7. There are no results of phase compositions of analysed particles. Only EDS measurements were performed.
Answer:
Following text has been added on page 9 and a neu image containing analysis of a couple of such segregations in the supplementary material – suppl Fig 3:
(rows 189-191): “Their chemical analyses by EDX showed a mixture of Ni, Mo, Ti, Nb, W with various concentrations most of them having a high concentration of Mo and Ni. Some examples of such segregations with the EDX analysis and the respective elemental quantifications are shown in suppl. Fig 3.”
(rows 201-204): “The chemical analysis of the EDX spectra in Fig. 3f) #2 shows following elements: Si – 0.24 wt.%, Mo – 4.09 wt.%, Cr – 5.01 wt.%, Mn - 0.34 wt.%, Fe – 69.02 wt.%, Co – 7.17 wt.%, Ni – 11.94 wt.%, Cu – 1.27 wt.% (systematical error of the EDX quantification in SEM (Oxford Si(Li) detector) is ~10%).”
8. What does it mean “martensitic refined microstructure”?
Answer:
We mean that the martensitic laths are quite small in size (150 nm to 1 µm). The text has been changed to: “refined microstructure”.
9. The EDS results of matrix should be added to the Fig. 5. A discussion should be carried out regarding differences in elements occurring in particles and matrix. Due to the specifics EDS method, the results of EDS shown in Fig. 5 come from matrix, as well.
Answer:
Yes, it is true. The EDX spectra (#1, #2 and #3) contain also signal from the matrix. For comparison an additional supplementary Figure 4 has been added in the supplementary material. This figure contains EDX spectra from the matrix and from a retained segregation with the respective elemental quantification.
10. Where are the results of NbC phase composition? There are no electron diffraction patterns.
Answer:
Following sentences have been added to the main text on page 8
“Chemical composition of the matrix as extracted from the EDX spectrum image that also serve as source for the EDX maps shows concentrations of the major elements like Fe: 77.31 ± 13.0 wt.%, Cr: 16.26 ± 1.9 wt.%, Ni: 3.37 ± 0.6 wt.% and Cu: 3.06 ± 0.5 wt.%. The Nb rich particles contain: C: 5.6 ± 0.3 wt.%, Nb: 39.64 ± 6.1 wt.% and the contribution from the matrix underneath: Fe: 37.22 ± 6.2 wt.%, Cr: 9.8 ± 1.5 wt.%, Ni: 2.68 ± 0.5 wt.% and Cu: 5.09 ± 0.8 wt.%. The Ni intermetallic phase contain: Ni: 11.02 ± 1.2 wt.%; with the contribution from the matrix being: Fe: 71.13 ± 12.2 wt.%, Cr: 14.91 ± 1.8 wt.% and Cu: 3.69 ± 0.6 wt.%. After normalization to the matrix and analysis of the FFT image calculated from the high-resolution STEM image presented in Suppl. Fig. 5, we found the Ni phase to be a µ - type (A6B7) phase.”
A line scan over one NbC particle is shown in Fig. 6 #3 (y-axis displays the wt.%). For such particles with nanometer sizes only the high-resolution imaging or nano-diffraction line scans (or mapping) would deliver reliable information. We did not perform nano-diffraction measurements since the first goal of our investigation at this stage was to prove the chemical stability of the nano-oxides and nano-carbides formed during 3D printing at higher temperatures. A following project will deal with the analysis of eventual metastable secondary phases containing Cu, Cr, Ni and Nb.
11. “Moreover, we also observed the precipitation of Cr23C6…”. Where are these results?
The nucleation of the M23C6 particles happened at higher temperatures above (620°C). Their formation is acknowledged in the literature [22, 23], as a phase that can form at higher temperatures but in a very low phase fraction. On the C- and Cr- EDX maps at 700°C in Fig. 11 the formation CrxCy particle can be observed. The chemical analysis shown in the line scan presented in Suppl. Fig. 8 corresponds to the Cr23C6 phase. We also suppose that we observed its formation also due to the surface diffusion the carbon atoms supplied by the fixing material of the FIB cut (mixture of Pt and C).
The text on page 10 has been reorganized Suppl Fig 8 and following sentences have been added:
“A particle containing both Carbon and Chromium can be observed on the related EDX maps. The chemical analysis shown in the line scan presented in Suppl. Fig. 8 corresponds to the Cr23C6 phase.”
12. The manuscript includes errors. For instance: “cupper segregation”.
The correction has been done.
13. The “mechanical properties of the final additively manufactured parts” are mentioned in the conclusions, but neither one of them applies to it.
We performed nanoindentation investigations to measure the Vickers hardness (HV1) (Fig. 2) of the stepwise samples in cross section (YZ plane).
14. Conclusions should highlight the new findings of the paper in comparison to the state of the art.
The conclusions have been improved.
Reviewer 3 Report
In this work, in situ high temperature TEM analysis was conducted on additive manufactured maraging steel via selective laser melting. The experiments were conducted in a good standard and the results were clearly presented. The reviewer would like to recommend it for publication with just a few minor issues to address.
- Figure 1. The powder looks hardly a spherical shape. Can you check again if this is recycled powder or as-received powder. Can you also explain the defects in the powder, i.e., the interlamellar crack-shape pore.
- Figure 2. There is no error bar in the hardness measurement. It was mentioned the hardness was measured by nano-indentation. can you please provide more details on the method? Did you measure on the surface or cross-section? You need to conduct a statistical analysis of the data as the difference is not very significant.
- Figure 3. What is exactly figure g and i? Can you just delete these two and explain it in the text?
- Figure 8. It looks like a lot of changes at temperatures over 850C. Why didn't you just increase the temperature more slowly and capture more detailed information between 850 to 950C?
- Figure 9. Again, oxidation occurred very rapidly at above 700 C. Why didn't you apply smaller increments between 700 and 760?
Author Response
Thank you very much for the review and for the valuable comments that have helped us to greatly improve this paper. We addressed all points and provided the error bars in Fig. 2 as well as explanations and the corresponding changes in text. An additional SEM image of the powder was introduced in the supplementary material.
- Figure 1. The powder looks hardly a spherical shape. Can you check again if this is recycled powder or as-received powder. Can you also explain the defects in the powder, i.e., the interlamellar crack-shape pore.
Answer:
An overview SEM image of the as-received powder was introduced in the supplementary material as Suppl. Fig. 1. The images in Fig. 1 are from a chosen irregular grain in the as-received powder. The irregular grain has a grain core and accretionary forms on top of it. The apparent “interlamellar crack-shape pore” it actually the interface between the grain body and the accretionary form named envelope in the main text. The formation of the accretionary form is probably due to the collision of two powder grains one of them being not entirely solidified at the moment of collision. It can also be linked with the thin amorphous carbon and copper layers on the surface of the solidified grain that acts as a “glue” for the envelope. More details about the powder is given in [20].
2. Figure 2. There is no error bar in the hardness measurement. It was mentioned the hardness was measured by nano-indentation. can you please provide more details on the method? Did you measure on the surface or cross-section? You need to conduct a statistical analysis of the data as the difference is not very significant.
Answer:
The error bars (statistical deviations) have been displayed in figure 2. Method details have been included in the text: page 3 rows 129 – 135. The samples were measured in cross section, for each step of the plate three measurements have been performed.
3. Figure 3. What is exactly figure g and i? Can you just delete these two and explain it in the text?
Answer:
Figures g and i represent the EBSD measurement of the retained austenite grains that have dimension between 280nm and 700nm, therefore they appear as very small (could also be wrongly interpreted as noise) at this magnification. In order to inquire the authenticity of this phase we replaced the image i and j with other EBSD images at higher magnification.
4. Figure 8. It looks like a lot of changes at temperatures over 850C. Why didn't you just increase the temperature more slowly and capture more detailed information between 850 to 950C?
Answer:
On one hand, we acquired images at 800, 850, 900 and 950°C and observed a dramatic deterioration of the thin lamella after 850°C. On the other hand, the STA measurements showed the second exothermic peak to have a maximum centered around 750°C heaving the decreasing slope ending at 900°C, therefore we expected the main coarsening process to slow down after 850°C.
5. Figure 9. Again, oxidation occurred very rapidly at above 700 C. Why didn't you apply smaller increments between 700 and 760?
Answer:
The second exothermic peak in the STA measurements exhibit a fast-increasing slope between 640° and 700°C and we payed attention to this temperature region since here the important nucleation of Ni and Cr precipitates happened. Above 700°C the decrease started and we took images at 740 and 760°C. The fast oxidation of the lamella (surface oxygen diffuses into the matrix) is probably favorized by the vacancies produced in the matrix by Cr and Ni diffusion while precipitates form and grow.
Round 2
Reviewer 2 Report
The paper is strongly improved. It is still necesarry to move the results included in the supplementary to the manuscript.
Author Response
Thank you very much for your suggestion!
Considering their importance, we moved following images to the main paper:
- Fig. 2 in the main paper as Fig. 1d and e
- Fig. 5 in the main paper as Fig. 6c
- Fig. 8 in the main paper to Fig. 11.
The results and discussion chapters have been correspondingly adapted and the corrections in the main text have been made with tracking. English language and style were improved.